# Physico-Chemical Study of the Possibility of Utilization of Coal Ash by Processing as Secondary Raw Materials to Obtain a Composite Cement Clinker

Bekkeldi Muratov [1], Alexandr Kolesnikov [1,*], Shermakhan Shapalov [1], Samal Syrlybekkyzy [2,*], Irina Volokitina [3], Dana Zhunisbekova [4], Gulchehra Takibayeva [4], Farida Nurbaeva [2], Taslima Aubakirova [1], Lazzat Nurshakhanova [5], Akmaral Koishina [5], Leila Seidaliyeva [2], Andrey Volokitin [6], Aizhan Izbassar [7] and Igor Panarin [8,*]

1 Department of Life Safety and Environmental Protection, M. Auezov South Kazakhstan University, Shymkent 160012, Kazakhstan
2 Department of Ecology and Geology, Sh. Yesenov Caspian University of Technology and Engineering, Aktau 130002, Kazakhstan
3 Department of Metallurgy and Mining, Rudny Industrial Institute, Rudny 111500, Kazakhstan
4 Department of Higher Mathematics and Physics for Technical Specialties, M. Auezov South Kazakhstan University, Shymkent 160012, Kazakhstan
5 Department of Petrochemical Engineering, Sh. Yesenov Caspian University of Technology and Engineering, Aktau 130002, Kazakhstan
6 Department of Metal Forming, Karaganda State Industrial University, Temirtau 101400, Kazakhstan
7 Department of Construction Engineering, Sh. Yesenov Caspian University of Technology and Engineering, Aktau 130002, Kazakhstan
8 Polytechnic Institute, Far Eastern Federal University, 690922 Vladivostok, Russia
* Correspondence: kas164@yandex.kz (A.K.); samal.syrlybekkyzy@yu.edu.kz (S.S.); panarin.ii@dvfu.ru (I.P.); Tel.: +7-7052566897 (A.K.)

**Abstract:** A significant amount of energy waste has accumulated in the world, in particular, large-tonnage fine ash from central heating stations (coal ash), which can negatively affect the natural environment and the health of the population. However, at the same time, due to its chemical composition, this waste can be disposed of by complex processing as a secondary mineral component, thus reducing the anthropogenic load on the natural environment. This article presents a physico-chemical study of coal ash for its further use as a secondary mineral component, in particular, a component of a raw mixture with limestone to produce a composite Portland cement clinker. Coal ash and limestone were subjected to granulometric, chemical, differential thermal, scanning electron microscopy, elemental chemical and X-ray structural analyses, as well as modeling to assess the possibility of optimizing the raw material and mineralogical composition of the composite Portland cement clinker. During the research, the chemical and elemental compositions of the coal ash and limestone were determined and SEM images of the coal ash were obtained; it was found that 68.04% of the coal ash was represented by the fraction with granules <0.16 mm. Using X-ray diffraction analysis, the main limestone minerals were identified, which were represented by calcite and silica. Based on the results of mathematical modeling of the utilization of coal ash from a thermal power plant by processing with limestone, a two-component raw material mixture containing 23.66% fly ash and 76.34% limestone was optimized and the optimal mineralogical composition of the composite Portland cement clinker was determined. Utilization of coal ash by processing as a secondary raw material can be carried out at almost any ash storage facility anywhere in the world, taking into account the chemical composition of the processed ash. It was found that the replacement of natural raw materials with man-made raw materials in the form of coal ash contributed to a reduction in fuel consumption for firing (kg of conventional fuel) by 13.76% and a decrease in the thermal effect of clinker formation by 5.063%.

**Keywords:** composite material; raw mineral; coal ash; limestone; modeling; utilization; mix; composite; cement industry; environmental engineering; industrial ecology

## 1. Introduction

It is urgently necessary to concentrate on employing efficient sources of renewable energies as a long-term solution to the problem of reducing emissions of greenhouse gases and the incidental environmental consequences of energy production [1–3]. Many nations, especially those with plentiful supplies of this renewable resource, are seeing an increase in biomass use for heat and power generation [4–6]. Globally, the number of biomass power plants is rapidly rising in an effort to meet the ambitious goals of reducing both our reliance on burning fossil fuels and greenhouse gas emissions, as well as a strategy to more sustainably manage our forests and prevent forest fires [7,8]. However, the substantial volume of ash created during biomass combustion for energy production is a drawback [9–11]. During the thermochemical conversion of biomass into energy, two different types of ash are formed: ash of lower origin (BA), which is a material removed from the bottom of the furnace, and fly ash (FA), which consists of low-density dust particles trapped in a gaseous state and collected, in particular, in the equipment for heat recovery after the furnace and flue gas purification system [9,10,12–14]. Both BA and FA must be removed and handled like hard waste, conforming to the European Commission's List of Wastes [15]. The amount of biomass ash produced annually around the world is increasing quickly (about 500 million tons per year), and the majority of the matter (about 70%) is currently being disposed of in landfills [16,17]. Therefore, there is a pressing need to identify stable ash management options in order to decrease the costs and unintended ecological consequences of landfill dumping [16,18,19]. The "end-of-waste" plan of the European Union toward a closed economy is consistent with the goals of valorizing biomass ash [20].

Utilization of coal ash is relevant and occupies a priority place among promising solutions in many countries of the world. Currently, only 10–15% of waste from thermal power plants is used in various industries, but the potential for its use is much wider [21–23]. The annual amount of waste generated in developed countries is up to 15 tons per person per year, up to 50–100 tons in countries with raw material economies [21], and about 60 tons in Kazakhstan [24]. The main share of waste in Kazakhstan falls on industrial areas and is represented by overburden rock from the mining industry, tailings from enrichment processes, and waste from heat electric power plants (TPP), in particular ash and slag [24–29].

Technogenic waste from TPP and hydroelectric PP includes ash, slag, and waste gases. The output of waste in the type of ash and ash slag depends on the type of fuel; 10–15% for brown coal, 30–40% for stone coal. In the Republic of Kazakhstan, the annual output of ash and ash slag waste from coal combustion is about 19 million tons, and today more than 300 million tons of waste have been accumulated in ash dumps. Over the past 50–60 years of thermal power plant operations in Kazakhstan and around the world, a huge amount of ash waste has been formed (Figure 1), which requires close attention in order to improve the environmental situation in the country and solve production problems associated with obtaining certain types of raw materials.

Consequently, this waste needs comprehensive disposal and recycling [24,25,30,31]. Among industrial wastes, the leader in terms of volume is coal ash waste from the combustion of solid fuel (various types of coal, peat, and shale) [32]. During the activities of TPP using coal as the main fuel, millions of tons of ash waste are generated annually on the territory of Kazakhstan, occupying huge areas of more than 100,000 hectares, and their maintenance requires significant financial investment, which contributes to an increase in the costs of energy carriers. This waste has a negative impact by polluting the natural environment, contributes to endangering the health of the population, and poses a threat to the fauna and flora of nearby areas [33]. Of particular danger are ash dumps near water basins (rivers and lakes), which are particularly dangerous due to the potential for dam breaks. As the amount of ash and slag materials increases, the area of territories allocated for ash dumping also increases, which decreases the amount of land intended for growing crops and grazing [34]. Looking at such practices from ecological and economic point of views, in addition to causing damage to the soil, groundwater, and clogging

the territory, natural resources that have already been extracted and processed once are irretrievably lost [35,36].

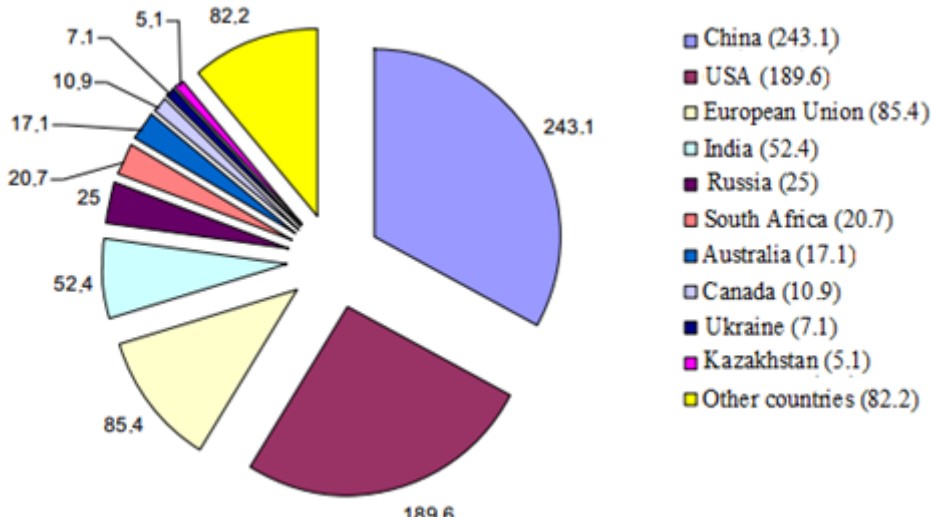

**Figure 1.** Accumulated ash waste in the countries of the world. Annotation: The numbers in the figure are measured in million tons.

For the sale of waste in the form of ash and various slags as raw compounds for various industries, and for its safe use and storage by various branches of industrial production in the national economy, it is necessary to have basic information on its properties and characteristics, such as its physico-chemical-mineralogical properties and composition as well as its environmental parameters (radiation and the presence of toxic properties) [37]. By its chemical composition, waste in the form of ash is directly dependent on the mineralogical and chemical structures of the fuel of a particular coal deposit [35]. One of the directions of ash waste disposal is its use in the production of porous aggregates for light concrete. Additionally, fine aggregate can be replaced with ash. As large aggregates, crushed stone obtained from fuel slag, agglomerate based on ash, burnt and unbaked gravel, and alumina-containing expanded clay are currently mainly used [34,38,39]. Fuel slag and ash are the raw compounds for the production of artificial porous aggregate as sinter. Using conventional technology, sinter is obtained in the form of crushed stone. Technologies for the production of sinter gravel from ash, alumina expanded clay, and ash gravel have also been developed. Alumina expanded clay is obtained by swelling and sintering granules formed from a mixture of clay and ash in furnaces [40]. Technologies for the production of fired and non-fired ash gravel have been developed, allowing the use of almost any ash obtained from the combustion of various types of coals. The efficiency of introducing up to 20–30% ash instead of cement in the manufacture of concrete and mortar has been established. It is especially advisable to introduce ash in the concrete of hydraulic structures. Ash waste is used for the production of silicate bricks, reducing lime consumption by 10–50% and sand by 20–30%. Such bricks have lower density than the usual ones. Fuel ash and slag are applied as thinning and and burn-out additives in the manufacture of ceramics based on clay materials and as the main raw material in the production of ash ceramics [40]. Ash ceramics are characterized by high acid resistance, low abrasion resistance, as well as high chemical and thermal resistance [41].

Fused materials are obtained from fuel ash and slag, including slag pumice and wool. A number of high-temperature technological methods for obtaining mineral wool by electric melting in an arc furnace are known. The mineral wool obtained in this way is intended for thermal insulation of surfaces in the temperature interval of 890–1100 °C. At the same time, it is also possible to obtain various kinds of glasses, a wide variety of architectural and construction products, and in particular, tiles for facing purposes [42,43]. Waste in the form

of ash is used as backfill in the building of roads and in building foundations in preparation for asphalt concrete floors. Ash is also used as filler for the production of mastics from rolled roofing materials [23,34]. For example, it is possible to produce concrete from the alumina obtained from the silicon ash of Ekibastuz coal and cement from the waste of alumina production; the concrete obtained has sufficiently high performance characteristics and gains good strength within one day. Waste from this industry can be used to prepare coal workings with the possibility of further disposal by recycling [34,44].

The use of ash slag in agriculture improves a number of soil indicators, particularly the agrophysical properties of the soil, which contributes to the replenishment of its micro- and macroelemental compositions, improves porosity, and neutralizes acidity. However, it is necessary to take into account the dangers of ash slag, such as radiation indicators, changes in water migration, general sanitary translocation, and toxicological indicators. Despite the obvious benefits and prospects for the widespread use of ash waste, the volume of its use in developing countries does not exceed 10%, and the disposal of accumulated ash requires solving a set of issues [45–48], from the development of technical conditions for its use, to technological lines for its processing, transport, loading, and unloading, to changing the psychology of business executives in relation to secondary mineral resources [23,34]. Research in the area of creating materials-saving technologies for processing waste from thermal energy plants has been a trend in the development of new scientific methods for several years, both in the world at large and in Kazakhstan [49].

In the coming decades, the world will not abandon coal power, and it will retain an important role in Europe and around the world. Today, 80% of energy in Kazakhstan is generated at coal-fired electrical stations. Therefore, the main energy sources in the country in the near future will be thermal power plants. One of them is the Balkhash thermal power plant located in Balkhash, Karaganda region (Figure 2), which is designed to cover the shortage of electricity in the southern region of Kazakhstan. For its operation, in all likelihood, cheap Ekibastuz coal will be used, which is characterized by high ash content; with gross excavation, it reaches 60%. That is, when burning 1 kg of coal, 600 g of ash is emitted. With large volumes of coal burning, a huge amount of solid particles entering the hall is released into the atmosphere [50]. Ash-based materials can be employed as nutritional supplements as well as pH correctors (ash pH is often >12) [51–54]. The use of ash in fertilizing products is now being discussed, as well as processing and product standards. However, some types of ash formed from biomass can also include significant amounts of toxic elements (TE), depending on their source [55–57]. To avoid any detrimental effects on the environment or human health, the ash of the CHP plant should be evaluated for its physico-chemical properties and elemental content prior to recycling [7,8,58].

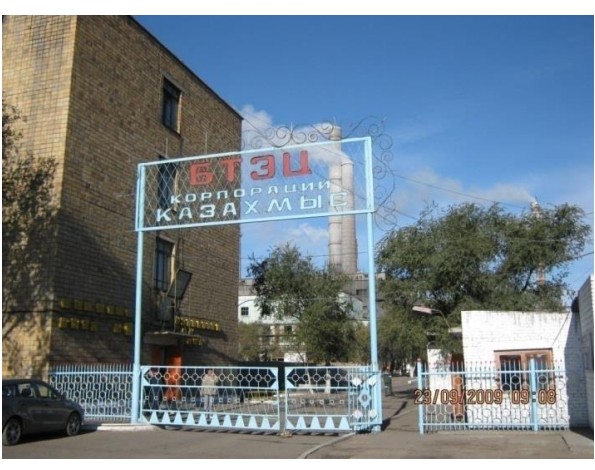

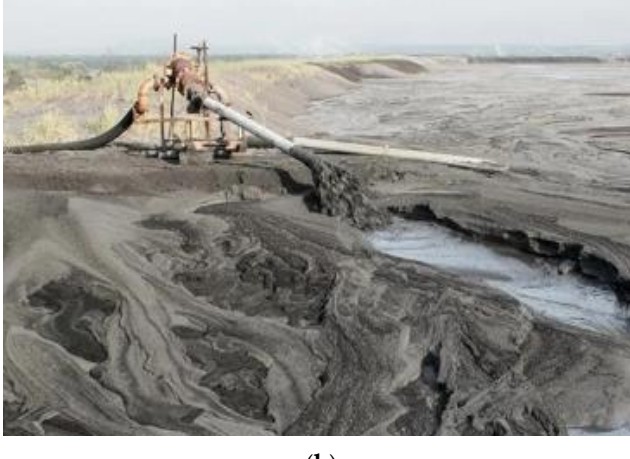

(**a**)  (**b**)

**Figure 2.** General view of the Balkhash power plant directorate (**a**), and general view of the accumulated ash (**b**).

Thus, the work directed at decreasing the energy and financial expenses associated with the extraction of raw mineral compounds by attracting technogenic raw compounds during the utilization and processing of waste, with a concomitant reduction in the anthropogenic load on the region and the living population, has novelty and relevance.

The huge amount of ash formed from burning coal to generate electric energy contributes to harmful effects on the environment of the region and the population. In particular, according to the Bureau of National Statistics of the Agency for Strategic Planning and Reforms of the Republic of Kazakhstan, there has been a stable number of identified patients with respiratory diseases in the region over the past 5 years. This dependence is associated with the presence of ash dumps, since the ash is in pulverized form and contains 42–53% silicon oxide as well as various silicates. Prolonged exposure to ash dust contributes to the activation of phagocytosis (absorption of harmful particles), in which macrophages play the main role. When macrophages absorb dust particles, reactive oxygen species are formed and pro-inflammatory mediators (arachidonic acid metabolites and cytokines) are released. In addition, the fibrogenic dust particles themselves activate inflammatory cytokines. In particular, the roles of cytokines such as tumor necrosis factor and interleukin-1 have been confirmed in the development of silicosis.

Atmospheric air is one of the main environments of the human habitat, and the health of the human body, the level of physical development, reproductive capability, susceptibility to diseases, and life expectancy largely depend on its quality.

However, air pollution still poses a significant threat to human health worldwide. According to the WHO's assessment of the burden of air pollution-related deaths, more than two million premature deaths are associated annually with the effects of urban and indoor air pollution, which is caused by the burning of solid fuels. Today, about 5 million residents of Kazakhstan live in conditions with polluted atmospheric air, while at least 2 million live in conditions with extremely high levels of pollution [44,45,59].

Based on a review of the methods of ash formation, processing, and disposal, the following conclusions can be drawn. In Kazakhstan, in particular, super-acidic ash waste continues to accumulate annually at landfills of the CHP, which has not found application in industry due to the complexity of processing. In addition, none of the available technologies allow mass processing of ash slag but only partially use it, and its utilization and involvement in recycling does not exceed 10%. In this regard, work aimed at exploring the possibility of recycling large-capacity waste from the energy industry, such coal-fired power plants, is relevant and of scientific interest to the global community. The aim of this research was to perform a physico-chemical investigation of the ash of the Balkhash thermal plant for its utilization as a secondary raw product, with modeling and optimization of the raw mixture and mineralogical content for the production of Portland cement clinker.

## 2. Materials and Methods

In this work, samples of ash from the ash dump of the Balkhash (Figure 3) thermal power plant (Balkhash, Karaganda region, Kazakhstan) and limestone from the Kiik deposit (Agadyrsky district of Karaganda region, Kazakhstan; 0.8 km southeast of the Kiik railway station, 170–180 km from Balkhash) (Figure 4) were subjected to physico-chemical studies. The limestone deposit is dated to the Kiik anticline of the Akbastau synclinorium (Figure 5).

The determination of the moisture content of the coal ash was carried out as follows. A 1 g sample was placed in a box pre-dried to a constant weight, which was placed in an oven heated to a $105 \pm 5\,°C$, and the sample was dried for 1.5–2 h, after which it was cooled in a desiccator and weighed on an analytical scale providing a random measurement error of no more than $\pm 0.01\%$ of the weight of the sample being weighed. After drying, the sample was weighed and the moisture content of coal ash was defined. The granulometric structure of the coal ash was determined by sieving.

The scanning electron microscopy method of analysis was used in the research. It is known that microscopic analysis is used for direct or indirect investigation for a wide variety of processes. It is most often used to study the shape and size of crystals, the

process of crystal growth and deconstruction, the detection of minerals by gauging their optical constants, the establishment of some crystal chemical features of the crystal structure (habitus, cleavage, fracturing, zonality, the presence of inclusions, porosity, etc. etc.), phase transformations in substances, diffusion processes, etc. If it is possible to prepare high-quality micro-preparations, microscopic analysis enables controlling individual stages of any process [60,61]. The scanning electron microscopy analysis was carried out using a JSM-6490LV scanning electron microscope.

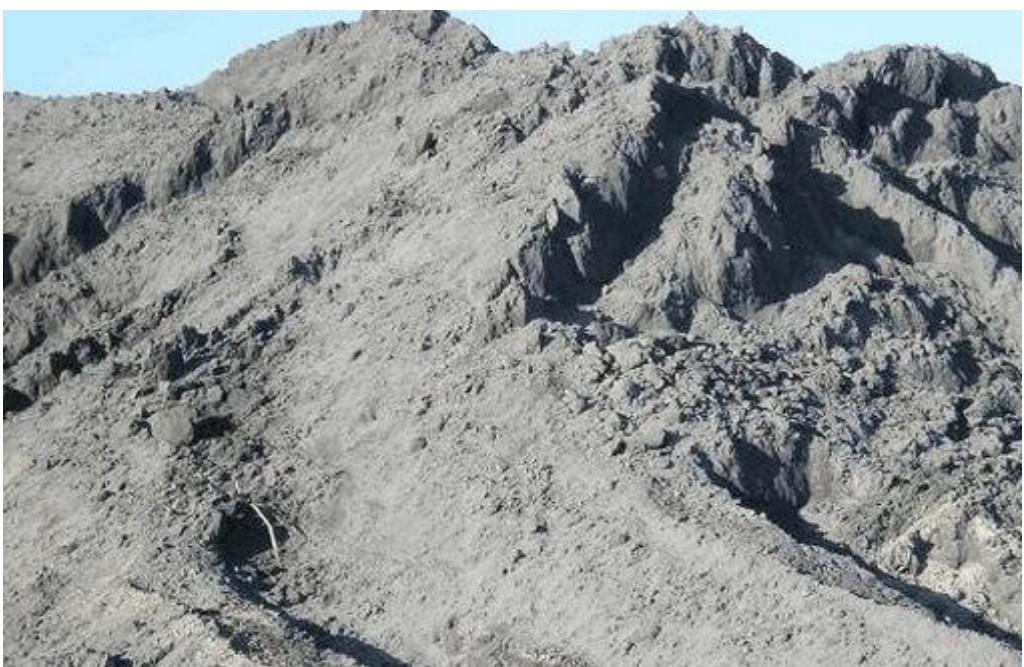

**Figure 3.** Ash dump of Balkhash CHP.

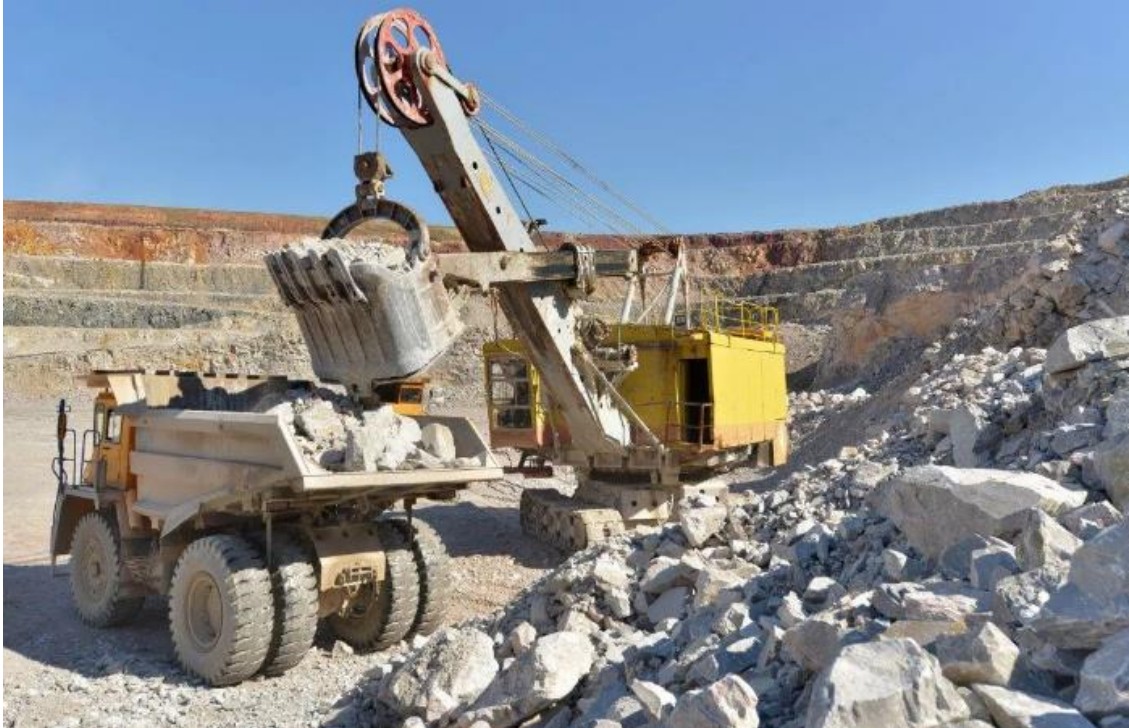

**Figure 4.** Limestone at the deposit.

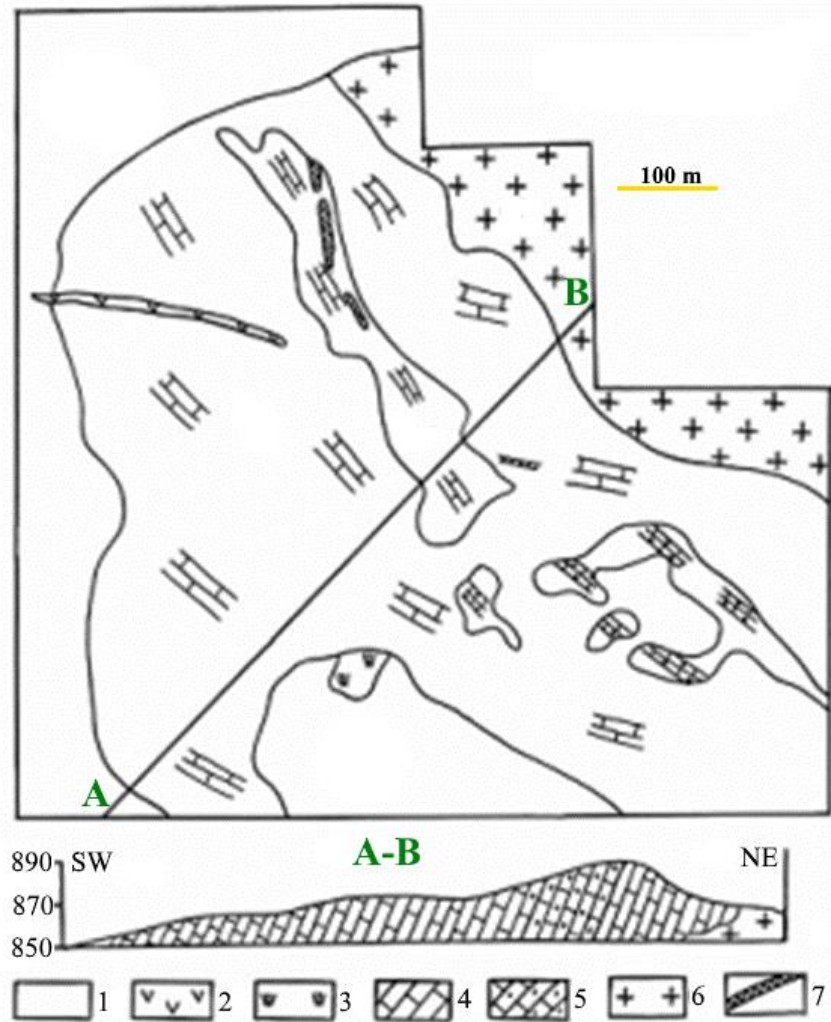

**Figure 5.** Limestone from Kiik deposit [30]. 1—Cenozoic deposits (soil and vegetation layer, loam); 2—quartz and felsic porphyry dykes; 3—quartzites; 4, 5—limestone of the lower-Middle Ordovician: 4—white and gray, 5—red-brown; 6—granodiorites of the lower Carboniferous; 7—quartz veins.

X-ray diffraction (XRD) analysis is a reliable experimental method for obtaining data on the structure and dynamics of the crystal lattice of solids under conditions of external influences such as pressure and temperature. This has a number of important benefits. For instance, the solid itself is investigated in an unaltered state, and the result of the analysis is a direct determination of the materials or its components. X-rays examine the crystal, i.e., the compound itself, but in the case of polymorphic bodies, X-rays make it possible to distinguish individual modifications peculiar to this substance. To study a substance, a very small amount of a substance is required, which is not destroyed during the analytical operation [62]. A DRON-3 X–ray diffractometer was used to perform the XRD analysis.

Compared to other methods, differential thermal analysis (DTA) is simpler and allows results to be obtained more quickly. In this case, the analysis was carried out using a Q-1500D derivatograph (Hungary) to simultaneously and automatically obtain temperature and differential heating curves and integral and differential weight loss curves. The weight of the material ranged from 0.2 to 10 g. The heating rate in the electric furnace was 0.5–20 °C per min. The maximum heating temperatures were 150, 300, 600, 900, and 1000 °C. A six-channel universal MV recorder served as the recording device, which provided the ability to register the results depending on time or temperature [63].

Modeling of the possibility of obtaining cement clinker in the process of utilizing coal ash was carried out using the "ROCS" software application (Belgorod, Russia), which was

specially developed by scientists of Belgorod State Technological University named after V.G. Shukhov for modeling, computations, and optimization of raw mixtures.

## 3. Results and Discussion

The humidity of the coal ash was determined by the difference in mass, which in our case was 1.34%. During the analysis of the physico-chemical properties of the technogenic ash of the thermal power plant and limestone, the granulometric composition of the coal ash was established. The results of the coal ash granulometric analysis are shown in Figure 6.

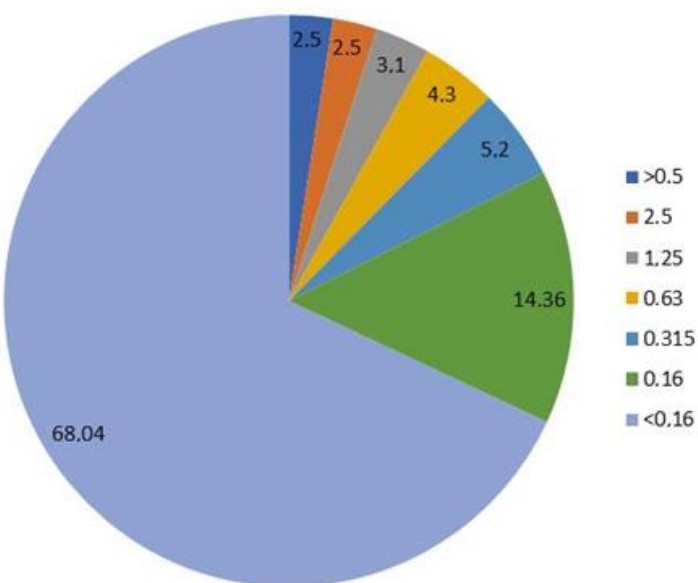

**Figure 6.** Granulometric composition of the coal ash. Annotation: The numbers in the figure are measured in (%).

During the granulometric analysis (Figure 6), it was found that the 68.04% of the coal ash was represented by the fraction with granules <0.16 mm. The analysis of the granulometric composition using scanning electron microscopy showed that 65% of the particles were represented by sizes from 10 to 70 μm. The size and morphology of the ash particles are shown in Figure 7.

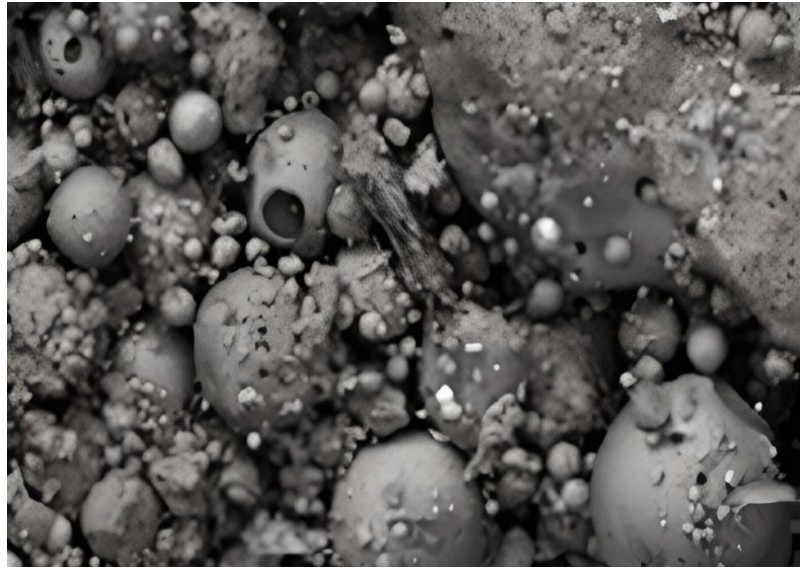

**Figure 7.** SEM image of the coal ash (__________ x500).

The micrograph of Figure 8 shows SEM image (**a**) and elemental analysis (**b**) of the ash from TPP.

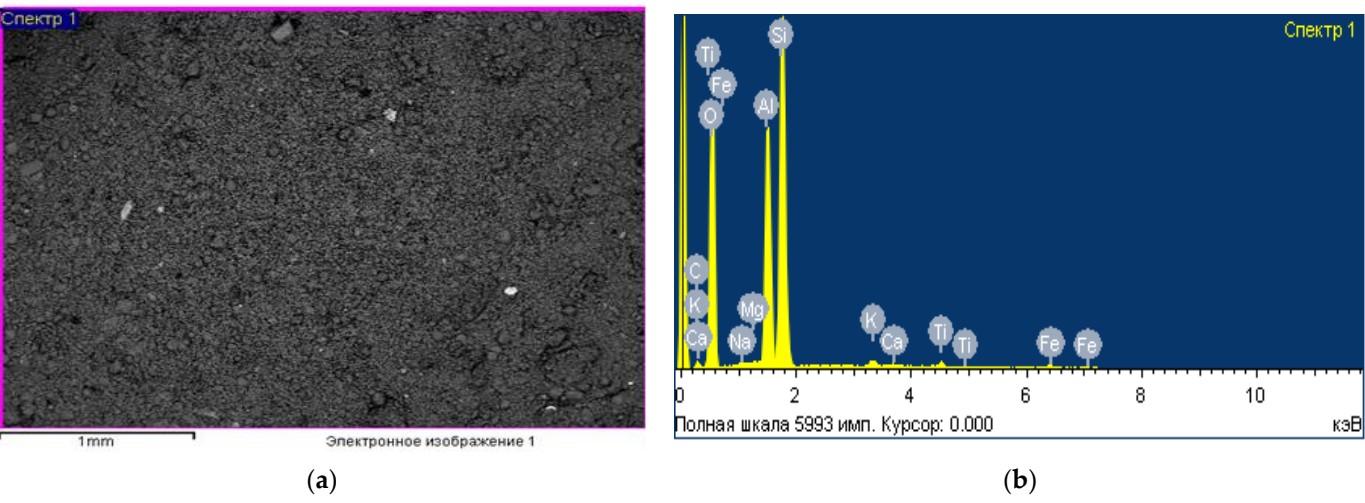

(**a**)                (**b**)

**Figure 8.** SEM image (**a**) and elemental analysis (**b**) of the ash from TPP. C—3.18%; O—51.98%; Na—0.25%; Mg—0.11%; Al—13.74%; Si—25.28%; K—0.60%; Ca—0.21%; Ti—0.82%; Fe—2.62%.

Based on the analysis of the ash, it was established that the ash was represented by silicon, aluminum, and iron oxides by a total of 94.2%, which made it possible to use it as a raw substance component to replace clay and iron-containing components in the production of cement clinker. This would contribute to reducing both the technogenic load on the surroundings and the population of the region and the costs of obtaining cement clinker by eliminating expensive operations for the development and maintenance of deposits of loess clays and iron ores and also excluding a whole chain of expensive operations to prepare the coal ash for grinding in a ball mill. The limestone from the Kiik deposit was subjected to XRD analysis using a DRON-3 apparatus (Figure 9).

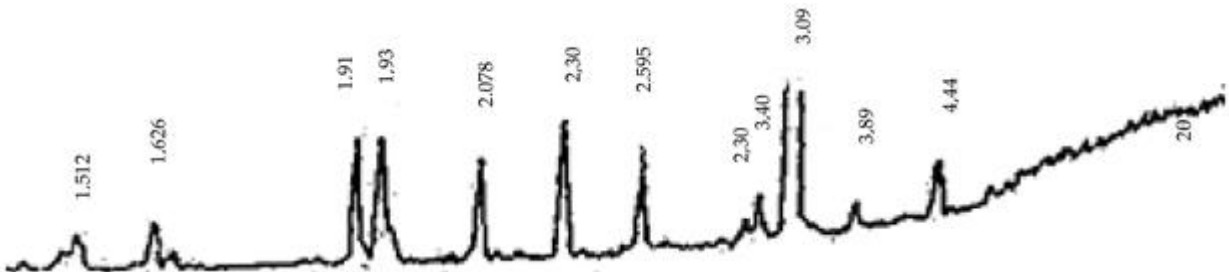

**Figure 9.** XRD pattern of the limestone.

In accordance with the results of the XRD analysis of limestone, Figure 9 shows that the limestone sample consisted mainly of calcite (d = 3.89; 3.09; 2.30; 2.078; 1.93; 1.910; 1.626; 1.512A) with admixtures of quartz (d = 4.44; 3.40; 2.595; 2.30A). This result was confirmed by a similar study [64]. The results of the differential thermal analysis of the limestone are shown in Figure 10.

The DTA curve of the limestone from the Kiik deposit (Figure 10) showed a deep endothermic effect with a maximum at 850 °C, indicating the decomposition of calcium carbonate. It followed from the figure that the calcite decomposition process began at a temperature of 650 °C. Physico-chemical studies have shown that limestone meets the technical requirements for its application as a raw component for the manufacturing of cement clinker [65]. Thus, based on the conducted physico-chemical studies of the technogenic waste of the energy industry—ash from the thermal power plant and raw

mineral material in the form of limestone from the Kiik deposit—the chemical compositions were established, which are shown in Table 1.

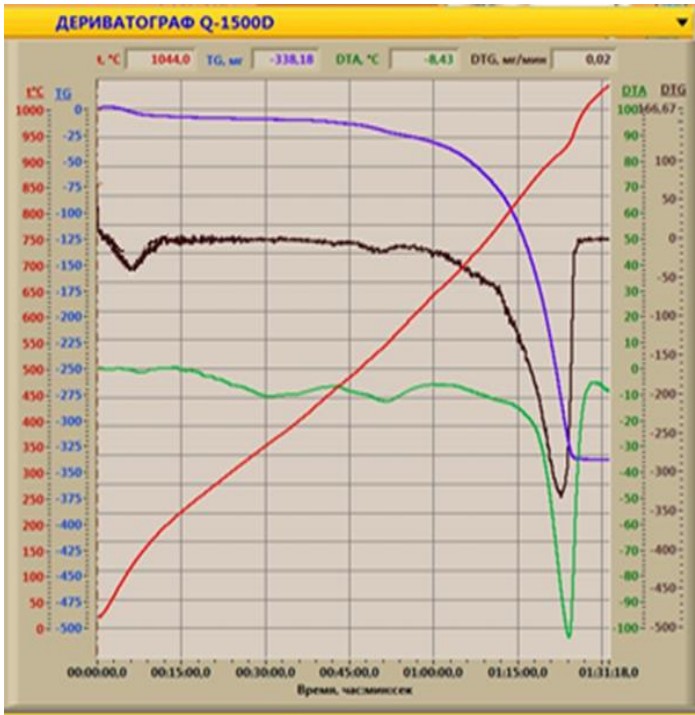

**Figure 10.** Differential thermal analysis of limestone.

**Table 1.** Chemical composition of technogenic waste—ash and limestone from the Kiik deposit.

| Chemical Compound | Material, % | |
| :---: | :---: | :---: |
| | Limestone | Coal Ash |
| $SiO_2$ | 1.07 | 54.80 |
| $Al_2O_3$ | 0.93 | 17.34 |
| $Fe_2O_3$ | 0.69 | 8.37 |
| CaO | 54.32 | 8.79 |
| MgO | 0.75 | 0.89 |
| Loss on ignition | 37.63 | 9.81 |
| Other | 4.1 | - |

Based on the chemical compositions of the coal ash and limestone, a simulation was carried out to optimize the raw mixture (coal ash and limestone) and the mineralogical composition of cement clinker depending on the saturation coefficient using the ROCS software application. The optimization results are shown in Table 2.

Based on the results obtained from the modeling carried out using the ROCS software application, it followed from Table 2 that it was possible to obtain cement clinker from coal ash and limestone that met the requirements. Thus, clinker could be produced from the raw mixture in the following composition at KN = 0.9: 23.66% coal ash of the thermal power plant and 76.34% limestone. Further, the mineral structure of clinker was: $C_3S$, 53.12%; $C_2S$, 17.23%; $C_3A$, 12.34%; $C_4AF$, 11.06%; and MgO, 1.14%. The technological parameters of the firing process were as follows: temperature, 1450 °C; silica module, 1.88; alumina module, 1.92; thermal effect of clinker formation (FEC), 336.7 kcal/kg; fuel consumption for firing, kg of conventional fuel ($G_{fuel}$), 188.7 fuel/t of cl.

Based on the conducted research, it was obvious that man-made waste such coal ash could be used as a secondary raw mineral material according to its chemical, elemental, and granulometric composition. The present coal ash was capable of acting as silica-, alumina-,

and iron-containing raw material in certain ratios with limestone in the raw mixture for the production of clinker by the method of high-temperature synthesis. The proposed method of utilization of coal ash could contribute to reducing its anthropogenic impact on public health and the environment, since ash can have a negative impact not only in the area of the ash dump but also significantly far beyond the territories of ash storage and warehousing. It is known that the process of dusting due to the influence of wind can lead to contamination of the soil, surface, and groundwater with toxic compounds. In particular, investigators found that ash components were detected in the superficial stratum of soil at a distance of several kilometers to several hundred kilometers from ash storage facilities, depending on the granulometric composition [64]. Additionally, the concentration of ash elements turned out to be increased in growing grasses and harvested forage studied near ash dumps and storage facilities. The toxicity and significant harm of coal ash to public health has been evidenced by many studies conducted by scientists, confirming the negative impact of ash on the health of people, especially those living in the zone of a thermal power plant and those with ash storage facilities in the immediate vicinity.

**Table 2.** The results of modeling the chemical and mineralogical compositions of the raw charge (mixture) and clinker with a saturation coefficient (SC) equal to 0.90.

| Chemical Composition of the Raw Charge (Mixture) and Clinker | | | | | | | |
|---|---|---|---|---|---|---|---|
| | $SiO_2$ | $Al_2O_3$ | $Fe_2O_3$ | CaO | MgO | Loss on Ignition | Other |
| Raw mixture | 13.78 | 4.81 | 2.51 | 43.55 | 0.78 | 31.05 | 3.52 |
| Clinker | 19.99 | 6.98 | 3.64 | 63.16 | 1.14 | - | 5.1 |
| Modules | | | | | | Raw mixture | Clinker |
| SC (lime saturation coefficient) | | | | | | 0.9 | 0.9 |
| n (silica module) | | | | | | 1.88 | 1.88 |
| p (alumina module) | | | | | | 1.92 | 1.92 |
| FEC (thermal effect of clinker formation, kcal/kg) | | | | | | - | 336.7 |
| $G_{fuel}$ (fuel consumption for firing, kg of conventional fuel/t of cl.) | | | | | | - | 188.7 |
| Mineralogical composition | | | | | | | |
| Minerals | $C_3S$ | | $C_2S$ | $C_3A$ | $C_4AF$ | $CaSO_4$ | MgO |
| Mas.% | 53.12 | | 17.23 | 12.34 | 11.06 | 0 | 1.14 |
| Content of components | | | | | | | |
| Materials | | | Raw mixture | | | Clinker | |
| | | | kg/kg cl | | % | % | |
| Limestone | | | 1.107 | | 76.34% | 69.05% | |
| Coal ash | | | 0.343 | | 23.66% | 30.95% | |
| Total | | | 1.45 | | 100.00% | 100.00% | |

Thus, researchers [65] found that the population living at a distance of 1000 to 2000 m from thermal power plants and ash storage facilities had a higher proportion of various diseases, in particular, upper respiratory tract infections, bronchitis, bronchial asthma, etc. According to the age groups of the population, children and persons over 40 years of age were more susceptible to such diseases in comparison with the population living in a favorable environment [66]. Such studies contribute to solving the problem of recycling the accumulated ash from heat energy plants, which contributes daily to the deterioration of the ecological status of the areas in which they are located around the world.

The substitution of CHP ash for traditional components of cement clinker production (such as loess clay and low-grade iron ore [67]) would reduce fuel consumption for firing, kg of conventional fuel ($G_{fuel}$), by 13.76%, reduce the thermal effect of clinker formation (FEC) by 5.063%, and, accordingly, reduce the firing temperature by 50–70 °C.

Thus, the results obtained during the research support the use of coal ash as a secondary raw mineral product for the manufacture of clinker, which complements the previously performed investigations of a number of researchers [63–69], and have all the

necessary indications for industrial testing and implementation of this method to obtain cement clinker by ash disposal at cement plants.

## 4. Conclusions

Based on the physico-chemical study results of the technogenic waste (coal ash) from the thermal power plant and limestone, with the modeling of the optimization of the raw mixture and the subsequent production of cement clinker, the following conclusions can be drawn. In particular, it was established that:

− Based on its chemical and elemental composition, the coal ash contained <95% aluminum, calcium, silicon, and iron oxide compounds, the existence of which contributed to the fact that coal ash acted as a secondary raw material for chemical production, in particular, a cheap clay-ferruginous component of the raw material mixture for the production of clinker;

− Based on the results of the raw mixture optimization modeling, the optimal ratios of coal ash (23.66%) and limestone (76.34%) were established at $k_n$ = 0.9;

− During the simulation of coal ash utilization as a raw material component of a raw mixture with limestone, the technological parameters of firing of the raw mixture at a saturation coefficient of 0.9 and temperature of 1450 °C were determined: in particular, silica module, 1.88; alumina module, 1.92; thermal effect of clinker formation (fec), 336.7 kcal/kg; fuel consumption for firing, kg of conventional fuel ($g_{fuel}$), 188.7 fuel/t of cl.;

− As a result of modeling, the optimal mineral structure of clinker was determined, which was presented by the following minerals: $C_3S$, 53.12%; $C_2S$, 17.23%; $C_3A$, 12.34%; $C_4AF$, 11.06%; and MgO, 1.14%;

− The replacement of traditional raw materials in the form of loess clay and iron ore in the raw material mixture for clinker production with waste in the form of ash from the Balkhash thermal power plant will contribute to a decrease in fuel consumption for firing, kg of conventional fuel ($G_{fuel}$), by 13.76%, a decrease in the thermal effect of clinker formation (FEC) by 5.063%, and co-responsibly contribute to a decrease in the firing temperature by 50–70 °C, thereby reducing the cost of the process, ash disposal, and the anthropogenic impact on the environment and the population, while improving sanitary standards and having positive social, economic, and environmental effects.

**Author Contributions:** Writing—review and editing, B.M., A.K. (Alexandr Kolesnikov), and I.P.; methodology, S.S. (Shermakhan Shapalov), S.S. (Samal Syrlybekkyzy); formal analysis, D.Z., G.T., and I.V.; investigation, F.N., T.A., and L.N.; supervision, A.K. (Akmaral Koishina) and L.S.; data curation, A.V.; resources, A.I.; validation, A.K. (Alexandr Kolesnikov). All authors have read and agreed to the published version of the manuscript.

**Funding:** The study is funded by the authors.

**Institutional Review Board Statement:** Not applicable.

**Informed Consent Statement:** Not applicable.

**Data Availability Statement:** Not applicable.

**Acknowledgments:** The authors acknowledge the support given and facilities provided by the Department of Life Safety and Environmental Protection, M. Auezov South Kazakhstan University, Kazakhstan.

**Conflicts of Interest:** The authors declare no conflict of interest.

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
