# Peer review of "Physico-Chemical Study of the Possibility of Utilization of Coal Ash by Processing as Secondary Raw Materials to Obtain a Composite Cement Clinker"

_jcs, doi:10.3390/jcs7060234_

Round 1

Reviewer 1 Report

This paper describes physico-chemical studies of coal ash for use as a secondary mineral component, where coal ash and limestone were subjected to granulometric, chemical, differential thermal, scanning electron microscopy, elemental chemical and X-ray structural analyses. The modelling of the possibility of optimizing the raw material composition and mineralogical composition of the composite- Portland cement clinker was conducted. The authors found that the replacement of natural raw materials with man-made raw materials in the form of coal ash contributes to a reduction in fuel consumption for firing by 13.76%, and a decrease in the thermal effect of clinker formation by 5.063%. The paper is interesting and could be accepted. 

The authors are recommended to read the text thoroughly and proofread the spellings and typos.

Author Response

Good afternoon, dear Reviewer!

Thank you for your support, I wish you creative success, health and all the best!

Sincerely, the Authors!

Reviewer 2 Report

The work concerns the physicochemical study of the possibility of utilization of coal ash by processing as secondary raw materials to obtain a composite cement clinker.

After carefully reading the work, I have comments:

1. The problem described in the work is interesting, but the authors should provide novelty.

2. The introduction is too long.

3. The drawings are hard to read.

4. The discussion should be more interesting in my opinion. There are too few examples of research by other scientists.

5. The first conclusion: "The coal ash has a fine granulometric fraction of 68.04% and, accordingly, is easily carried by the wind for many kilometers, polluting the environment and harming the health of the population" is known information. In my opinion, this should not be a conclusion.

6. There are a few minor editorial errors in the work - marked in yellow.

It is worth noting that the authors cite a lot of new literature sources.

Author Response

Good afternoon, dear Reviewer!

Thank you for your valuable comments and suggestions that contributed to the improvement of our article!

1The novelty lies in the fact that when using ash, mineral raw materials are replaced by man-made, which entails resource conservation with simultaneous utilization of ash and reduction of anthropogenic impact on the region of the location of TPP ash waste. And in turn, in the medium-long term, it contributes to improving the economic indicators of the implementation of this method. This is indicated in the article.

2 In the introduction, we tried to note all the prerequisites for the disposal of ash by recycling and gave the necessary arguments.

3 There is a small problem with the quality of the drawing, but it cannot be improved since it is given by the equipment on which the study was conducted.

4 We have tried to consider and give a number of examples similar and closer to our method and technology.

5 Agree with you. Eliminated.

6 Corrected and changed.

I wish you creative success, health and all the best!

Sincerely, the Authors!

Round 2

Reviewer 1 Report

All the comments have been addressed.

Reviewer 2 Report

All my comments have been taken into account.